# Predictors of Adherence to Cancer-Related mHealth Apps in Cancer Patients Undergoing Oncological or Follow-Up Treatment—A Scoping Review

**DOI:** 10.3390/ijerph192013689

**Published:** 2022-10-21

**Authors:** Christoph Armbruster, Marie Knaub, Erik Farin-Glattacker, Rieka von der Warth

**Affiliations:** Section of Health Care Research and Rehabilitation Research (SEVERA), Medical Center—University of Freiburg, Faculty of Medicine, University of Freiburg, 79106 Freiburg, Germany

**Keywords:** mHealth, usage, adherence, influencing predictors, cancer patients

## Abstract

mHealth interventions in cancer care are being increasingly applied in various settings. Nevertheless, there is a phenomenon wherein individuals show different usage patterns, which could affect the effectiveness of the intervention. In general, it is important to know the predictors of app adherence and usage patterns to improve the design and content (i.e., tailoring). The aim of this scoping review was to provide an overview of predictors of adherence to cancer-related mHealth apps in cancer patients. A systematic literature search was conducted in March 2021 in the electronic databases PubMed, CINAHL, and PsychINFO without limitation in year, focusing on cancer patients undergoing oncological or follow-up treatment using mHealth apps. The initial database search yielded a total of *N* = 8035 records. After title, abstract, and full-text screening, 10 articles met inclusion criteria. Studies were published between 2013 and 2020. Studies focused on children and adolescents (2/10) as well as adults (8/10). The predictors identified could be categorized into sociodemographic variables, cancer-related factors and others. This study provides an initial insight into relevant predictors of app adherence in cancer patients. However, no clear predictor of increased app adherence was found. Further research of usage patterns is therefore needed so that mHealth interventions can be tailored during development.

## 1. Introduction

Globally, cancer is one of the most common causes of death. In 2018, the prevalence of cancer cases was 18.1 million and 9.6 million died from it [1]. With rising numbers, cases worldwide are expected to reach 29.7 million in 2040 [1]. This leads to a number of challenges in several areas; cancer puts a great economic burden on nations and their health care systems [2]. For instance, the cost of cancer care was estimated to be around $208.9 billion in 2020 in the US [3] and €199 billion in 2018 in the EU [4]. In addition, cancer also puts a high financial burden on the patients themselves. According to Yabroff et al. [5], the net annual out-of-pocket costs in the US (i.e., medical services and prescription drugs) among adults aged 65 years and older across all cancer types were $2.443 during treatment and $4.271 in the terminal stage per patient. In addition, treatment of and living with cancer were shown to have a negative impact on patients health-related quality of life [6,7]. Physical symptoms such as pain or treatment-related side effects (e.g., insomnia, fatigue) are common consequences [8,9]. Furthermore, patients report increased depressive symptoms or uncertainties about their future [10,11]. Health care systems are therefore presented with the challenge of finding a way to address these various needs [12].

Mobile technologies play an integral role in today’s society. Mobile technologies are defined as technologies that can be used almost anywhere [13]. Most often, mobile technologies include internet-enabled devices such as smartphones, tablets, or watches [13]. For instance the worldwide number of smartphone users in 2020 was around 3.6 billion people [14]. Furthermore, in 2017 a total of 1.14 billion people owned a tablet [15]. This trend led to a huge increase in the development of mobile health (mHealth) apps. mHealth apps can be considered a part of electronic health (eHealth) which offer health services though several different functionalities and designs [16,17]. With regard to oncological care, a recent review identified 123 mHealth apps which are available in the two most important marketplaces (i.e., Apple iTunes and Google Play) [18] and there is growing evidence that these mHealth apps are effective [19,20,21]. Typical areas of usage in cancer are disease management support (e.g., symptom monitoring, management of side effects, medication reminder and dosing, access to health information), support of healthy behavior (e.g., healthy diet, increased physical activity), or the connection with other patients (e.g., social support through peers) [9,16,22]. The Covid-19 pandemic, has led to eHealth strategies becoming even more important in cancer care. According to the recommendations of Curigliano et al. [23], cancer patients should be offered eHealth strategies for exactly these specific usage areas (i.e., in the area of disease management and the support of healthy behaviors).

Nevertheless, there is a phenomenon wherein individuals show different usage patterns in eHealth interventions (i.e., no usage, stop using after a time period, only use a specific function and not all of the features, usage as recommended) [24,25,26]. In this matter, the term “adherence” comes into focus. As regards eHealth interventions, Donkin et al. [25] defined adherence as “the degree to which the user followed the program as it was designed”. It is assumed that the effectiveness of eHealth interventions is greater when patients show high usage. On the other hand, a different usage pattern (e.g., early completers) may not be directly related to a lack of interest or failure to achieve treatment goals [27]. Overall, it is important to evaluate predictors of adherence and usage patterns, which may help to improve the design and content (tailoring) and thus the usage behavior [28]. For example, a recent systematic review investigated predictors of adherence to online psychological interventions. Characteristics such as female gender, higher treatment expectancy, sufficient time and a tailored intervention led to greater adherence, while age and baseline symptom status showed contrasting results [29]. In the context of mHealth apps for cancer treatment support, there is a lack of such specific information to date or research has focused only on a specific age group and included a broader range of eHealth interventions [30]. Insights therefore need to be gained into which predictors influence the adherence of cancer-related apps. Thus, the aim of this scoping review was to answer the research question: what are predictors of adherence to cancer-related mHealth apps in cancer patients? For this purpose, we will provide a novel overview of possible influencing factors.

## 2. Materials and Methods

We conducted this scoping review using the preferred reporting items for systematic reviews and meta-analysis extension for scoping reviews (PRISMA-ScR) guidelines [31].

### 2.1. Search Strategy

We performed a systematic literature search in the electronic databases PubMed, CINAHL, and PsychINFO. We selected these databases as they represent the most content in the field of oncological care. To test the search strategy, a first pilot search (C.A.; R.W.) was applied. For this, we identified the keywords in relevant articles [32,33,34,35,36,37] and searched for additional synonyms. Additionally, we screened the results, refined, and subsequently adapted the strategy for the main search. One author (C.A.) searched each database without limitation in year or article type in March 2021. The search strategy used in the main search was based on the following search terms: cancer, radiotherapy, onco*, neoplasms, mobile applications, telemedicine, mhealth, adhere* adopt*, accept*, engage*, usab*, usage, eval* and feasab*. The search strategy was based on the PICO(S) scheme and used in combination with Boolean operators and truncation. Appendix A presents all of the used search terms in their combinations and there corresponding number of records. All duplicates were removed before the study selection process. Data management was carried out using Citavi [38].

### 2.2. Selection of Studies

M.K. and R.W. screened the titles and abstracts of the found articles independently. As we expected, that the measuring predictors of adherence would not be the primary outcome in most studies, we decided to include trials, feasibility studies, and observational studies on cancer-related apps for full-text screening when they mentioned the correct population and intervention (mHealth App) within the abstract. Potential conflicts (rate: 4.9%) were resolved in discussion with C.A., after the screening was completed. C.A. and R.W. then independently screened the full-texts for eligibility. Again, potential conflicts (rate: 4.1%) were resolved in discussion after screening was completed. We used the software Rayyan [39] for the complete screening process. The full eligibility criteria for study inclusion in this scoping review can be seen in Box 1.

Box 1Eligibility criteria for study inclusion.**Inclusion Criteria:** 
(1)Populations of cancer patients undergoing oncological or follow-up treatment(2)All of the age groups in populations (e.g., children, adults, elderly)(3)mHealth app as a main part of the intervention with no specific treatment goal, but cancer related(4)Studies measured and reported at least one outcome on factors influencing app adherence(5)Peer-reviewed empirical studies (quantitative primary studies, mixed methods studies)(6)Studies in English or German
**Exclusion Criteria:** 
(1)Population only undergoing cancer prevention screening (secondary prevention)(2)Studies in which only medical staff or relatives were involved(3)Browser-based apps which can also be accessed via computer(4)Abstracts, case reports, study protocols, letters, and editorials


### 2.3. Data Extraction

Data extraction was carried out by C.A. and R.W. in a joint process until consensus was reached. For each study, we extracted publication information (i.e., authors; year of publication), population characteristics (i.e., gender, age, cancer type), information on the app (i.e., name, purpose of the app), study characteristics (i.e., sample size, duration of observation). Finally, we collected data on adherence measurement as well as our main measure: significant predictors of cancer-related app adherence (e.g., sociodemographic variables, cancer-related variables etc.).

### 2.4. Data Analysis

To analyze the found predictors, we decided to categorize them. In total, three domains could be built: sociodemographic variables, cancer-related factors and others.

## 3. Results

### 3.1. Study Selection

The database search yielded a total of *N* = 8035 records. After removing duplicates *n* = 4917 records remained for title and abstract screening. Title and abstract screening yielded *n* = 209 studies for full-text screening (4.25%). In the full-text screening, reasons for exclusion were: no predictors of app adherence reported, no smartphone app or tablet app, wrong population, study language not English or German, full-text not available. Furthermore, one study was excluded [40] as it reported the same data as another study included in this review [34]. Finally, 10 articles remained after title, abstract and full-text screening and were included in the scoping review. Figure 1 demonstrates the selection and screening process and the main reasons for exclusion due to the PRISMA-ScR guidelines [41].

### 3.2. Included Studies

Included studies were published between 2013 and 2020. Of the 10 included studies, 2 were conducted in Canada and 2 in the Republic of Korea. Of the remaining studies, one study was published in Switzerland, Spain, Sweden, China, Germany and USA respectively. This scoping review contains a total of *N* = 986 patients, ranging from *n* = 14 to *n* = 181. The duration of observations ranged from 2 to 48 weeks. The included studies had different designs (i.e., observational study [42,43,44], secondary data analysis [45,46], mixed-methods study [40,47,48], quasi-experimental study [49], and randomized trial [50]). Table 1 provides an overview of the included studies.

### 3.3. Population Characteristics

The population characteristics of the included studies are shown in Table 1. Two studies included children ranged from 8 to 18 years with different types of cancer (i.e., haematological cancer, sarcoma [42,48], such as central nervous system tumor, renal cell cancer, and others) [42]. In the remaining studies, cancer diagnosis were haematological cancer [50], sarcoma [43,50], melanoma [50], genitourinary cancer [50], gastrointestinal and colon cancer [34,43,50], lung cancer [34,43,50], breast cancer [34,43,44,45,46,47,50], prostate cancer [43,47], ovarian or cervical cancer [34], glioma [43,50], and others [43,50]. Two studies included patients after acute cancer treatment (i.e., at least six months after adjuvant therapy [49], and in follow-up care for patients with a history of cancer [43]). In five studies, patients were treated with different treatment approaches (i.e., radiotherapy [42,44,47], chemotherapy [42,46,50], surgery [42,44], stem cell transplant [42], targeted therapy or antihormonal therapy [44,50]. Three studies did not specify treatment approaches or treatment phase as inclusion or exclusion criteria for participants [34,45,48].

### 3.4. Intervention and App Characteristics

The intervention and app characteristics of the included studies can be seen in Table 1. All of the interventions were accessible via smartphones or tablets. The apps of the included studies followed various objectives: seven studies used apps for reporting symptoms [42,43,46,47,48,50], reporting mental health [44,45] or reporting quality of life [43,50]. Four studies used apps to monitor or promote healthy behavior (e.g., physical activity, healthy diet, mindfulness and relaxation) [34,44,49,50]. In addition, four apps aimed to improve illness knowledge, and social support, or offered the possibility to create communities [44,46,47,49,50], while one app additionally provided the schedule of personalized medication dosing [50]. In order to achieve this objectives, the apps have been designed with different functions or tools: The Pain Squad App included a multidimensional pain diary for pain and treatment tracking, an automated function to alert the research team, and different gamification-elements to sustain user engagement [42,48]. Facial emoticon scales were used in the Pit-a-Pat app to collect three mental-health outcomes (i.e., anxiety, mood and sleep satisfaction). Additionally, the Patient Health Questionnaire-9 (PHQ-9) assessment [51] was carried out with the Pit-a-Pat app biweekly [45]. Mikolasek et al. [34] used a mindfulness and relaxation app, which contained mindfulness and relaxation exercises (i.e., mindfulness meditation, guided imagery, and progressive muscle relaxation) as well as a notification feature to promote daily exercise. The WalkOn app offered platforms for users with different approaches: tracking physical activity (i.e., daily steps, duration, distance) and sleep patterns supplemented with the integration of Fitbit [52], building communities and communicating with other app users, and viewing the number of daily steps taken by other users for self-motivation. The app also included self-reporting mental-health outcomes (i.e., anxiety, sleep, and emotion [45,53], distress (distress thermometer), the PHQ-9 assessment [51]) and notification functions [44]. Self-recorded physical activity regarding duration and intensity (Minnesota Leisure-time Physical Activity Questionnaire [54]) and diet behavior in respect of food and drink intake (with a dietary record questionnaire) were collected in the BENECA mHealth app too. This information was used for targeted recommendations on physical activity or diet via the app [49]. Buergy et al. [43] used the CAREONLINE app to obtain information on symptoms and quality of life. Another app for symptom tracking (daily on weekdays during treatment) was Interaktor. This app had an integrated reminder function to remind patients of their entries if they had not been carried out. Additionally, self-care recommendations were sent based on their entries (only included in the breast cancer version of the app) [47]. Greer et al. [50] used an app with a personalized medication dosing schedule, an adherence and symptom reporting module, educational resources and the integration of Fitbit [52] for tracking physical activity. The app also included reminders for the targeted oral cancer medication and push notifications for weekly reports. Zhu et al. [46] used the BSC app which had four forums to provide information and improve symptom management during treatment with chemotherapy: The Learning Forum provided evidence based information on breast cancer and related symptoms; the Discussion Forum provided an anonymous platform to communicate with peers and health care professionals; the Ask-the-Expert Forum provided health consultations within 24 h (if needed); and the Your Story Forum provided videos of encouraging stories to help manage the challenges of chemotherapy.

### 3.5. Measurement and Predictors of Cancer-Related App Adherence

Most frequently, app adherence was recorded via the number of usages in different dimensions (i.e., number of completed entries [48], expected vs. observed app entries [42], ratio of all of the answered daily questions or all of the received answers to all of the push notifications in a certain day [43], number of completed data collection or symptom reports [44,47,50], logging data or login frequency [46,49], free text messages sent [47], triggered alerts [47], views of self-care advice [47], and completed exercises per week [34]). Additionally, Zhu et al. [46] and Greer et al. [50] recorded the duration of use over an observation period of 12 weeks. Kim et al. [45] used a construct of three dimensions (activeness, timeliness and persistence) that clustered the level of adherence in two groups (high vs. low). Activeness was therefore calculated as the total number of days in which mental health ratings were carried out. Timeliness was captured by the total number of days immediately in mental health ratings. Persistence was considered using two variables: the total duration, measured with the number of two-week periods between the first and last day of mental health ratings and the total number of biweekly periods with reported ratings.

### 3.6. Predictors for App Adherence

We categorized the predictors influencing cancer-related app adherence into three domains: sociodemographic variables (e.g., age, gender, educational level, relationship status), cancer-related factors (e.g., symptoms, cancer stage, type of therapy) and others (e.g., presence of comorbidities, personality traits). The type of the measurement and the significant predictors of cancer-related app adherence is demonstrated by Table 2.

### 3.7. Sociodemographic Variables

Age was shown to be a relevant predictor in several studies. Crafoord et al. [47] showed that among breast cancer patients, the number of free text messages sent increased with higher age (*p* = 0.04). In the same study regarding prostate cancer patients higher age (*p* = 0.01) was also significant, albeit for the number of views on self-care advice [47]. Zhu et al. [46] showed that higher age was positively associated with the usage duration of the entire BSC app program (*p* = 0.003) and the Learning Forum (*p* = 0.008). In contrast Lozano-Lozano et al. [49] showed that higher age increased the risk of app attrition (*p* = 0.001). Chung et al. [44] showed that the number of days on which data collection was completed increased when patients were young, but only with the addition of other cancer-related predictors (*p* = 0.02). In terms of gender, one study showed that females had a better adherence to using the app continuously over time than men (*p* = 0.005) [34]. Regarding to the relationship status, another study showed that prostate cancer patients who were married or in a partnership reported their daily symptoms more often (*p* = 0.02) [47]. Two studies showed that the educational level was a significant predictor. As regards the prostate cancer patients, Crafoord et al. [47] showed that a higher educational level was significant with the total number of views on self-care advices (*p* = 0.04). Zhu et al. [46] showed that the higher educational level increased the usage duration of the entire BSC app program (*p* = 0.01), the usage duration of the Discussion Forum (*p* = 0.01), the login frequency of the Learning Forum (*p* = 0.01), and the login frequency of the Ask-the-Expert Forum (*p* = 0.02). The monthly family income was positively associated with similar dimensions (i.e., usage duration of the Learning Forum (*p* = 0.002), the login frequency of the entire BSC app program (*p* = 0.04), the login frequency of the Learning Forum (*p* = 0.04), and the login frequency of the Discussion Forum (*p* = 0.01) [46]. Regarding employment status, the same study reported a dichotomous association with the usage duration and the login frequency. On the one hand, this resulted in a higher login frequency of the entire BSC app program (*p* = 0.002), the Learning Forum (*p* = 0.001), Discussion Forum (*p* = 0.002) and Your Story Forum (*p* = 0.01), whilst on the other, it led to a lesser usage duration of the Ask-the-Expert Forum (*p* = 0.04) and the Your Story Forum (*p* = 0.03) [46]. In five studies, sociodemographic variables did not contribute to a significant increase in app use [42,43,45,48,50].

### 3.8. Cancer-Related Factors

Among children and adolescents, app use decreased due to pain in the last 12 h [42]. A targeted type of therapy (*p* = 0.009) or antihormonal therapy (*p* = 0.01) resulted in increased use among female breast cancer survivors [44]. Zhu et al. [46] did not find statistical differences with regard to cancer stage, types of surgery, or cycles of chemotherapy. Greer et al. [50] showed that the type of cancer had no significant influence in app use. Additionally, the initial treatment location (in-patient vs. out-patient) did not show any statistical difference [48].

### 3.9. Others

Two studies showed contrasting results on app adherence in the presence of comorbidities [44,47]. Chung et al. [44] operationalized the presence of comorbidities with a dichotomous variable while Crafoord et al. [47] used the Charlson Comorbidity Index [55]. In the study by Chung et al. [44], the presence of comorbidities led to an increase in app use. In the study by Crafoord et al. [47], prostate cancer patients with a higher comorbidity score showed less self-care advice views. Another study found no association between comorbidities and app adherence [46]. Cancer patients with higher depression values (operationalized through the HADS [56]) showed more continuous app use (*p* = 0.046) [34]. Patients with a higher score in openness to experience (operationalized through the NEO 5-Factor Inventory [57]) had a better adherence over time (*p* = 0.044) [34]. In addition, Mikolasek et al. [34] showed that patients with a higher score in resistance to change (operationalized through the Resistance to Change Scale [58]) had a better adherence in continuous app use (*p* = 0.03).

## 4. Discussion

### 4.1. Principal Findings

To our knowledge, this is the first scoping review that provide insights into the relationship between predictors and the adherence of cancer-related mHealth apps in cancer patients. We were able to include a total of 10 studies in this review and extract significant predictors for mHealth app adherence. We then categorized the predictors into three domains: sociodemographic variables, cancer-related factors, and others, with the most relevant predictors being found in the sociodemographic domain. Overall, we found no clear evidence in the predictors, but the following factors seem relevant and should be discussed further: age, employment status, education level and income, marital status, pain level, type of therapy, and comorbidities.

In relation to the sociodemographic variables, we found contrasting results in respect of age. While two studies reported an increase in usage by age [46,47], two did not [44,49]. According to previous studies, we would have expected that younger age was generally associated with higher adherence [59,60,61,62]. However, inconsistent results were also shown in a systematic review on online psychological interventions targeting psychological outcomes for a mental or physical health condition [29]. We would hypothesize that smartphones or tablets are now accepted in daily use not only by younger but also by older patients and can therefore be a promising option in the context of cancer care. In line with the literature [61,62,63], female patients showed better app adherence in this scoping review. In addition, female patients are more likely to prefer health apps related to self-healthcare [64], which was part of the intervention in the study by Mikolasek et al. [34]. Moreover, Venkatesh et al. [65] showed that the usage decisions of female are strongly influenced by the perceived ease of use that could be provided by the everyday usability of apps.

Only one study showed significant but mixed results in terms of employment status [46]. Employed female breast cancer patients showed more usage on specific app contents (i.e., in the Learning Forum and the Discussion Forum) which might be explained by the fact that working women want to maintain their ability to work and are most likely to benefit from this app content. In this context, Vaghefi and Tulu [60] reported that the presence of high motivation to achieve a specific health goal (e.g., weight loss or smoking cessation) in patients who have not had a chronic disease in the past is a key factor for the continued use of mHealth apps. We would assume that these findings might also be transferred to other goals (i.e., maintain the ability to work) and conditions, which could be another explanation for the increased app use by employed female breast cancer patients in our study.

Furthermore, previous research showed that people with a higher education level are more likely to search for health information in the digital world [61,64,66]. These findings were confirmed by 2 studies in our review, in which a higher level of education similarly contributes to a higher use of apps [46,47].

The finding by Zhu et al. [46] that cancer patients with a higher monthly family income showed more app use is consistent with the results of a prior study that investigated the use of health apps among low-income patients (e.g., with higher rates of obesity or chronic conditions [67]) accessing services at community health centers [68]. In addition, a lower income could lead to less use of information technologies related to mHealth [69]. Another explanation might be that a higher monthly family income correlates with a higher educational level and thus contributes to higher app use.

One study showed higher app use among patients who are married or in a partnership [47]. This finding is in line with a recently published study in which married patients with hypertension showed higher smartphone or tablet use to achieve their health related goals (e.g., quitting smoking, losing weight, or increasing physical activity) [62].

In terms of cancer-related factors, one study in our review reported that higher pain levels reduced app use concerns children and adolescents [42]. In this regard, the literature provides unclear evidence. Wang and Qi [64] showed that people with a lower self-rated health status are less likely to use mHealth apps. In contrast to this, people who consider themselves as more ill are more likely to use information technologies as a source for information [70]. Furthermore, the study by Stinson et al. [42] in our review is among children and adolescents, who are to be considered differently in respect of pain perception [71]. Regarding the type of therapy, one study reported that a targeted or antihormonal therapy led to higher app use while another study found no significant results in this matter [44,46].

In relation of the domain “others”, we found inconsistent results in the presence of comorbidities [44,47]. One explanation could be that the presence of comorbidities was assessed differently and thus led to different results. One other study reported higher app use in cancer patients with higher depression values [34]. In contrast to earlier findings, however, no evidence of depression or anxiety was detected to adherence in digital interventions [72,73,74]. In the same study, Mikolasek et al. [34] showed that higher values in different individual traits (i.e., openness to experience and resistance to change) increased app adherence. This is in line with a prior study, which had investigated the influence of individual traits on the use of mHealth apps in patients with diabetes [75].

### 4.2. Strengths and Limitations

To the best of our knowledge, this scoping review provides the first overview of significant predictors of app adherence in cancer patients. Thus, these results could contribute to the further development of tailored apps in cancer care, which remains a challenge. PRISMA guidelines (PRISMA-ScR) were rigorously followed in conducting this scoping review [29]. We applied a sensitive literature research, fitted it to our research question using a pilot search and searched in three databases. As we expected that most of the results relevant for this review were not the main topic within published articles, the search and inclusion criteria were chosen without initial strict limitations. However, due to the limited number of databases used for the search, and as the reporting of predictors of app adherence was often not the main scope of published articles, we cannot exclude that all of the important articles were included. However, our search and inclusion strategy was chosen to be sensitive, resulting in a high number of search results and inclusion to full-text screening (*n* = 209). Furthermore, the validity of our results is limited. For example, the mean age in most of the studies was less than 60 years, and 4 of 10 studies only included female participants [44,45,46,49], leading to a reduced generalizability for certain patient groups. Furthermore, the included studies were relatively heterogeneous in study population and study design (e.g. 2 studies included children and adolescents [42,48]). This also relates to the observational period, investigated apps and functions, and the operationalization of adherence and predictors. This might also be the reason why we were not able to identify certain predictors in some studies, for example in sociodemographic data.

Specifically in the context of the operationalizing adherence, Sieverink et al. [27] recommend that adherence measures do not only rely on the assumption of “the more use, the better”, but also to specify a justification for the threshold of intended use. In this way, it might also be possible to compare the level of adherence across different mHealth interventions [27]. To our knowledge, this has not yet been implemented, so in our scoping review we have equated the term “adherence” with the term “use” or “usage”.

## 5. Conclusions

In summary, we were able to provide a novel insight into the relevant predictors of app adherence in cancer patients. However, there remains relatively little evidence regarding this, and many studies of cancer-related mHealth interventions do not report on adherence and its predictors. This is also undermined by the fact that we found 146 studies which reported an evaluation of a mHealth intervention in cancer patients but did not report any further information on usage.

We would therefore encourage researchers to investigate usage patterns and predictors in a more cohesive way, so that the tailoring of mHealth interventions can be conducted during development. In addition, researcher should consider the recommendations on operationalizing app adherence so that comparability is given.

## Figures and Tables

**Figure 1 ijerph-19-13689-f001:**
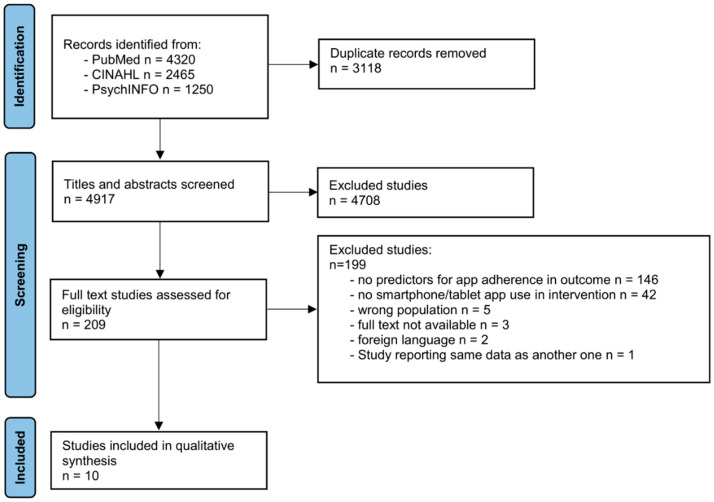
PRISMA flowchart.

**Table 1 ijerph-19-13689-t001:** Population, intervention and app characteristics of the included studies.

Study	Country	Population	Appname	Purpose
Stinson et al. (2013) [48]	Canada	Children and adolescents diagnosed with cancer (9–18 years)	Pain Squad App	Pain-Ratings
Stinson et al. (2015) [42]	Canada	Children and adolescents diagnosed with cancer (8–18 years)	Pain Squad App	Pain-Ratings
Kim et al. (2016) [45]	Republic of Korea	Breast cancer patients (Mean 44.35; SD 7.01)	Pit-a-Pat App	To collect several mental-health PROs of breast cancer patients
Mikolasek et al. (2018) [34]	Switzerland	Any cancer (age 18 or older)	N/S ^a^	Mindfulness and Relaxation
Chung et al. (2019) [44]	Republic of Korea	Female Breast Cancer Survivors (20–65 years)	WalkOn App (integrated with Fitbit)	Monitor physical activity and sleep patterns + creating communities
Lozano-Lozano et al. (2019) [49]	Spain	Breast cancer survivors between 30–75 years	BENECA App	Monitor and provide feedback on healthy eating and physical activity
Buergy et al. (2020) [43]	Germany	Patients of 60 years and older with a history of cancer	CAREONLINE	Symptom and quality of life reporting
Crafoord et al. (2020) [47]	Sweden	Patients with breast cancer or prostate cancer undergoing neoadjuvant chemotherapy and radiotherapy	Interaktor	Symptom reporting
Greer et al. (2020) [50]	Massachusetts, USA	Patients with diverse malignancies who were prescribed oral therapy for cancer (age 18 or older)	N/S ^a^ (integrated with Fitbit)	Personalized medication dosing schedule, adherence and symptom reporting module, educational resources for symptom management and other cancer-related topics, Fitbit integration for tracking physical activity
Zhu et al. (2020) [46]	China	Female Breast Cancer patients undergoing chemotherapy	BCS app program	Information and social support to improve symptom management during chemotherapy

^a^: N/S, not specified.

**Table 2 ijerph-19-13689-t002:** Study characteristics, type of the adherence measurement and significant predictors of the included studies.

Study	Sample Size	Duration of Observation	Adherence Measurement	Significant Predictors
Stinson et al. (2013) [48]	*N* = 14	2 weeks	Number of completed entries within the 2-week period	N/A ^a^
Stinson et al. (2015) [42]	Study 1—*N* = 92 Study 2—*N* = 14	Study 1 = 2 weeks Study 2 = 3 weeks	Expected vs. observed App entries	Pain within the past 12 h (−)
Kim et al. (2016) [45]	*N* = 85 (Dropout of *N* = 7)	48 weeks	Construct of several factors (activeness, timeliness, duration and persistence) that determine adherence level (low vs high level)	N/A ^a^
Mikolasek et al. (2018) [34]	*N* = 100 (Dropout of *N* = 46)	10 weeks	Number of completed exercises per week	Female gender (+) ^b^ Openness to experience operationalized using the NEO-FFI ^c^ (+) Depression *operationalized using the HADS* ^d^ (+) Resistance to change (+)
Chung et al. (2019) [44]	*N* = 160	6-month	Number of days data collection was complete	Low Age (+) ^e^ Comorbidities (+) ^e^ Antihormonal therapy (+) ^e^ Targeted therapy (+) ^c^
Lozano-Lozano et al. (2019) [49]	*N* = 80	8 weeks	Logging data	Low age (+)
Buergy et al. (2020) [43]	*N* = 54 (Dropout of *N* = 25)	4 months	Individual rate: Ratio of all answered daily questions to all push notifications Daily rate: Ratio of all received answers from different patients on a certain day to all push notifications	N/A ^a^
Crafoord et al. (2020) [47]	*N* = 149	Breast cancer = 18 weeks Prostate cancer = 9 weeks	Symptom report, triggered alerts, views of self-care advice, text-message use	Breast cancer: High Age (+) Prostate cancer: High age (+) High education level (+) High comorbidity score (−) Being married or cohabitating (+)
Greer et al. (2020) [50]	*N* = 181 (Mobile App *N* = 91 and Standard care *N* = 90 )	12 weeks	Minutes and days of app use, completed symptom reports	N/A ^a^
Zhu et al. (2020) [46]	*N* = 57	12 weeks	Usage duration, login frequency	High Age (+) High Education level (+) High Family income (+) Employment status (−/+) ^f^

^a^: N/A, not available. ^b^: Only gender was significant after multivariate cox proportional hazards regression. ^c^: NEO-FFI, NEO Five-Factor Inventory. ^d^: HADS, Hospital Anxiety and Depression Scale. ^e^: All predictors were significant only when clinical factors (type of therapy) were included in the regression model. ^f^: (−/+), Significant, but mixed results in terms of use of specific app content.

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
