# Peer review of "Predictors of Adherence to Cancer-Related mHealth Apps in Cancer Patients Undergoing Oncological or Follow-Up Treatment—A Scoping Review"

_ijerph, 2022, doi:10.3390/ijerph192013689_

Round 1

Reviewer 1 Report

As authors of this paper shown, this is the first scoping review that provide insights into the relationship of predictors and the adherence of cancer-related mHealth apps in cancer patients. Methodology was well described. In my opinion , the authors used a very limited number of databases to search for data , which is the limitation of this study.Please added this information in limitation part. 

Author Response

Dear Reviewer,

Reviewer 2 Report

This scoping review addresses the predictors of mHealth adherence in cancer patients. The results may allow to adjust the app’s design and content. I think it is an interesting article. However, I would like to make some suggestions.

 1.  Abstract: I suggest rewriting the eligibility criteria (For example: articles published from DATE to DATE, including patients with cancer undergoing oncological or follow-up treatment using mHealth apps).

 2. Introduction: I suggest including a research question.

 3. Materials and Methods:

- Is there any registration number or web address of the protocol for this scoping review?

- I suggest adding two items: “Main Measures: Predictors for app adherence” (sociodemographic variables, cancer-related factors, and others) and “Data Analysis”

 4. Results:

·     - I suggest adding the participant age range of all included studies.

·      - The following paragraph of “Study Characteristics” could be inserted in “Included studies”:    This scoping review contains a total of N = 986 patients, ranging from n = 14 to n = 181 within the included studies. The duration of observations ranged from 2 to 48 weeks.  The included studies had different designs (i.e., observational study [43,45,47], secondary data analysis [44,50], mixed-methods study [40,42,48], quasi-experimental study [46], and randomized trial [49]).”

 5. Discussion

·       -The authors could include the main results in the first paragraph.

·      - Did the included studies have many older participants, promoting difficulty for adherence to cancer-related mHealth apps? The presence of more younger patients may lead to limitations in the analysis of the results.

·     -Some included studies had only female participants which could be mentioned as a limitation for analysis data.

Author Response

Dear Reviewer,
